# Overactive bladder and bladder pain syndrome/interstitial cystitis in primary Sjögren's syndrome patients: A nationwide population-based study

Chun-Kang Lee[1], Ching-Pei Tsai[2], Tsai-Ling Liao[3,4], Wen-Nan Huang[5,6], Yi-Hsing Chen[1,5,6], Ching-Heng Lin[3,7,8,9☯], Yi-Ming Chen [iD] [3,4,5,6☯] *

1 Department of Medical Education, Taichung Veterans General Hospital, Taichung, Taiwan, 2 Department of Obstetrics, Gynecology & Women's Health, Taichung Veterans General Hospital, Taichung, Taiwan, 3 Department of Medical Research, Taichung Veterans General Hospital, Taichung, Taiwan, 4 Institute of Biomedical Science and Rong Hsing Research Center for Translational Medicine, National Chung Hsing University, Taichung, Taiwan, 5 Division of Allergy, Immunology and Rheumatology, Taichung Veterans General Hospital, Taichung, Taiwan, 6 Faculty of Medicine, National Yang-Ming University, Taipei, Taiwan, 7 Department of Public Health, Fu-Jen Catholic University, New Taipei City, Taiwan, 8 Department of Health Care Management, National Taipei University of Nursing and Health Sciences, Taipei, Taiwan, 9 Department of Industrial Engineering and Enterprise Information, Tunghai University, Taichung, Taiwan

☯ These authors contributed equally to this work.
* ymchen1@vghtc.gov.tw

**Data Availability Statement:** The National Health Institute Research Database (NHIRD) owns the minimal data and thus restricts the public

## Abstract

To investigate the risks of overactive bladder (OAB) and bladder pain syndrome/interstitial cystitis (BPS/IC) in primary Sjögren's syndrome (pSS) patients. A nationwide, population-based cohort study was conducted using data from Taiwan's National Health Insurance Research Database. From 2001 to 2010, participants with newly diagnosed pSS were recognized as the study group. In addition, a comparison cohort of non-pSS participants was matched for age, gender, and initial diagnosis date. Risks of developing OAB and BPS/IC in pSS patients of different age, sex, and various therapeutic strategies were calculated. Hazard ratios (HR) and a 95% confidence interval (CI) were analyzed by Cox proportional hazard model. In total, 11,526 pSS patients were recognized. The HRs of OAB and BPS/IC in pSS patients were 1.68 (95% C.I.: 1.48–1.91, $p<0.01$) and 2.34 (95% C.I.: 1.59–3.44, $p<0.01$), respectively. The risks of OAB and BPS/IC were significantly increased for pSS patients aged < 65 years (HR: 1.73 and 2.67), female patients (HR: 1.74 and 2.34), and patients requiring treatment for dry eyes and dry mouth (HR: 2.06 and 2.93). pSS patients exhibited an increased risk of OAB and BPS/IC. Female gender, younger age, and severe glandular dysfunction requiring treatments were potential risk factors.

## Introduction

Primary Sjögren's syndrome (pSS) is a systemic inflammatory autoimmune disease primarily affecting lacrimal and salivary glands. It is estimated to affect 0.9–6 per 1000 individuals, with

availability. NHIRD includes registry for beneficiaries, original claim data of inpatient and ambulatory care, prescriptions at pharmacies, registry for medical facilities and board-certified specialists from 2001 to 2013. The authors of the present study had no special access privileges in accessing data from NHIRD which other interested researchers would not have. This study utilized a subset of the NHIRD, the Longitudinal Health Insurance Database (LHID) of randomly selected 1,000,000 enrollees from the NHIRD. To access the database, the applicant must be a researcher / clinician from a university, research institute, or hospital, and the use of the data must be for research purposes only. All applications will be peer-reviewed to ensure the rationality of the use. Researchers must follow the Computer-Processed Personal Data Protection Law in Taiwan, and sign an agreement declaring no attempt to retrieve information violating the privacy of patients or health care providers. Find instructions at the following link: https://dep.mohw.gov.tw/DOS/np-2497-113.html.

**Funding:** YMC received a grant (TCVGH-1087312C) from Taichung Veterans General Hospital, Taiwan (https://www.vghtc.gov.tw/). The funders had no role in study design, data collection and analysis, decision to publish, or preparation of the manuscript.

**Competing interests:** The authors have declared that no competing interests exist.

female predominance and is typically diagnosed in the 4th and 5th decade of life.[1] Up to 98% of patients suffer from dry eyes or dry mouth. Female patients may also experience dyspareunia due to vaginal dryness. Approximately 50% of patients develop extraglandular manifestations, including musculoskeletal, hepatic, renal, pulmonary, and neurologic involvements.[2] In addition, bladder disorders may occur in pSS patients.[3]

Overactive bladder (OAB) is described as a syndrome with clinical manifestations of urinary urgency, frequency and nocturia. For lack of bladder infection or other overt pathology, OAB may also exhibit with or without urge urinary incontinence. Interstitial cystitis (IC), i.e. bladder pain syndrome (BPS), is a disease characterized by chronic bladder pain, increased urinary urgency as well as frequency. The overall prevalence of BPS/IC is 10.6 cases per $10^5$ patient-year (women: 45 per $10^5$ patient-year and men: 8 per $10^5$ patient-year).[4] Although functional somatic syndromes are closely associated with BPS/IC, the etiology of OAB and BPS/IC remains incompletely understood and is considered to be multifactorial.[5]

In 1993, van de Merwe was the first to report an association between pSS and BPS/IC.[6] Moreover, urinary dilatation may also occur in pSS patients with BPS/IC, leading to obstructive renal insufficiency.[7] A hospital-based study also showed that the prevalence of OAB symptoms was more frequently encountered in pSS patients.[3] It was proposed that autoantibodies binding to the $M_3$ muscarinic receptor ($M_3R$) of SS patients may cause exocrine dysfunction and lead to bladder detrusor smooth muscles contraction. Furthermore, acute urination symptoms and late inflammatory changes may be observed in the urinary bladder. [8] However, large-scale, population-based studies of pSS-related bladder disorders are lacking. This study aimed to investigate the association between pSS, OAB, and BPS/IC using a retrospective nationwide cohort.

## Material and methods

### Data source

A population-based retrospective cohort study was conducted using Taiwan's National Health Insurance Research Database (NHIRD), which includes registry for beneficiaries, original claim data of inpatient and ambulatory care, prescriptions at pharmacies, registry for medical facilities and board-certified specialists from 2001 to 2013. More than 98% of the citizens in Taiwan are included under the National Health Insurance program. All medical claims for outpatient and inpatient services are stored in the NHIRD. The Longitudinal Health Insurance Database (LHID), a subset of the NHIRD comprising medical claims of 1,000,000 systematically and randomly selected enrollees from the NHIRD, was used in this study. The Bureau of National Health Insurance (BNHI) insured the integrity and correctness of the claims databases. This study was approved by the Institutional Review Board of Taichung Veterans General Hospital, Taiwan (CE17178A). As the NHIRD contains de-identified and anonymous claim data opened to the academic community for epidemiological studies, informed consent from each participant was exempted.

### Participants

In this study, we enrolled a study group and a comparison group (Fig 1). All patients with the diagnosis of pSS by (International Classification of Diseases, 9th Revision, Clinical Modification [ICD9-CM] Code 710.2) during the period 2001–2010 were enrolled in the study cohort. The first outpatient visit with a ICD9-CM code for pSS in the same period was defined as the index date for the study cohort. In order to enroll new pSS subjects for analysis the hazard ratio of OAB and BPS/IC, those who had been diagnosed with pSS before January 1, 2001 were excluded. Subjects younger than 18 years or without sex data were excluded. The confirmation

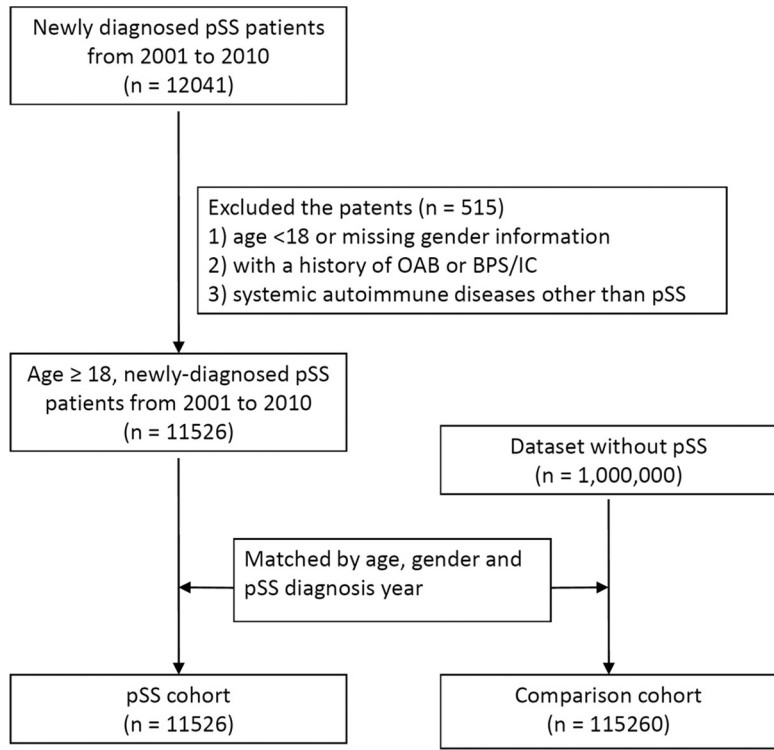

**Fig 1. Flow chart for study population selection.** pSS: Primary Sjögren syndrome; OAB: overactive bladder; BPS: bladder pain syndrome; IC: interstitial cystitis.

of pSS diagnosis was also guaranteed by the catastrophic illness certificates, which requires an evaluation by two independent rheumatologists. Participants with a prior diagnosis of either BPS/IC or OAB before the index date were excluded. Subjects with a systemic autoimmune disease diagnosis other than pSS during 2001–2010 were not included for further analysis. The study cohort included 11,526 newly diagnosed pSS subjects.

The comparison cohort without pSS was selected individually to match the pSS cohort (1:10) for age at disease onset, sex and year of index date. We excluded subjects with outpatient claim data less than 3 years after the index date. Totally, 115,260 patients without pSS were comprised of the comparison cohort. The study cohort and comparison cohort were followed up until December 31, 2013 or until either BPS/IC or OAB, occurred.

## Study outcomes

The occurrence of either BPS/IC or OAB was defined as the primary outcome. For BPS/IC (ICD9-CM code 595.1) and OAB (ICD0-CM code 596.51, 596.54, 596.55, 596.59, 788.31, 788.33, 788.34, 788.39), subjects should have at least three outpatient or one inpatient diagnosis. Subjects with OAB were also confirmed by records of treatment by flavoxate, imipramine, oxybutynin, propiverine, solifenacin, tolterodine, mirabegron or trospium for more than 30 days.

## Comorbidities and pharmacological treatment

Hypertension (ICD9-CM code 401–405), coronary artery disease (CAD, 410–414), type 2 diabetes mellitus (DM, 250), chronic pelvic pain (CPP, 625.9), chronic fatigue syndrome (CFS,

780.71), depression (296.2, 296.3, 296.5, 300.4, 309, 311), anxiety (300.0, 300.2, 300.3, 308.3,309.81), migraine (346), fibromyalgia (729.0, 729.1), irritable bowel syndrome (IBS, 564.1), asthma (493), sleep apnea (780.51,780.53,780.57), and hyperthyroidism (242) were identified as comorbid medical disorders.[9] Comorbidities were categorized if these diagnostic codes were found in ≧ 2 outpatient claims six months before and after the index date. Pharmacologic treatment for pSS was also included. Pilocarpine, cevimeline and saliva substitute were defined as treatment for dry mouth, while artificial tears were categorized as treatment for dry eyes.

## Statistical analysis

Independent t test was used to examine the unadjusted comparisons of categorical variables. Cox proportional hazard model was used to study hazard ratios (HR) of incident OAB and BPS/IC a with a 95% confidence interval (95% CI). Age, sex and comorbidities were adjusted for. Log-rank test was applied for the comparisons of the Kaplan–Meier survival analysis. All data were analyzed using SAS software version 9.1 (SAS System for Windows, SAS Institute, Cary, N.C., USA). A significance level was set at of $p < 0.05$.

## Results

### Demographic data

In total, our study included 11526 cases and 115260 controls (Table 1). The comorbidities of CAD, CPP, depression, anxiety, migraine, fibromyalgia, UTI, IBS, asthma, sleep apnea, and hyperthyroidism were significantly higher in the pSS cohort compared their counterpart.

### Risks of OAB and BPS/IC

Compared with the comparison cohort, pSS patients exhibited significantly increased risks of OAB (Table 2, adjusted hazard ratio, aHR: 1.68, 95% C.I.: 1.48–1.91, $p < 0.01$) and BPS/IC (aHR: 2.34, 95% C.I.: 1.59–3.44, $p < 0.01$). Fig 2 shows the disease-free survival curves of OAB and BPS/IC using the Kaplan–Meier method. The five-year OAB and BPS/IC-free survival rates for the pSS cohort were significantly lower compared to the control cohort ($p < 0.01$ by Log-Rank test, respectively).

We further analyzed the risks of OAB and BPS/IC between pSS patients and the control group by age and sex (Table 3). When compared with the non-pSS group, pSS patients aged < 65 years showed consistently higher risks of OAB (aHR: 1.73, 95% C.I.: 1.45–2.06, $p < 0.01$) and BPS/IC (aHR: 2.67, 95% C.I.: 1.73–4.12, $p < 0.01$). However, risk of BPS/IC in participants aged ≥65 years was similar in both groups. Moreover, female pSS patients exhibited increased risks of OAB (aHR: 1.74, 95% C.I.: 1.52–2.00, $p < 0.01$) and BPS/IC (aHR: 2.34, 95% C.I.: 1.59–3.44, $p < 0.01$) when compared with the non-pSS group. On the other hand, male participants did not show significantly increased risks of bladder irritation syndromes ($p = 0.09$).

### Comparison of risks of OAB and BPS/IC by pSS-specific treatments

To investigate whether pSS-specific treatments may be associated with bladder symptoms, we further analyzed risks of OAB and BPS/IC by various therapeutic strategies (Table 4). Interestingly, we found that pSS subjects with either treatment for dry eye or both treatment for dry eye and dry mouth had higher risks of bladder irritation symptoms. However, treatment of dry mouth alone was not significantly associated with risks of OAB and BPS/IC.

**Table 1. Comparisons of demographic data between pSS and comparison cohorts.**

|  | Comparison cohort N = 115260 | pSS cohort N = 11526 | p value |
|---|---|---|---|
| Age, year (SD) | 53.8 (14.2) | 53.8 (14.1) | 0.68 |
| Sex |  |  | >0.99 |
| Female | 102260 (88.7) | 10226 (88.7) |  |
| Male | 13000 (11.3) | 1300 (11.3) |  |
| Comorbidity |  |  |  |
| Hypertension | 38677 (33.6) | 3874 (33.6) | 0.91 |
| CAD | 19048 (16.5) | 2652 (23.0) | <0.01 |
| DM | 18136 (15.7) | 1752 (15.2) | 0.13 |
| CPP | 258 (0.22) | 49 (0.43) | <0.01 |
| CFS | 224 (0.19) | 33 (0.29) | 0.04 |
| Depression | 7449 (6.46) | 1792 (15.6) | <0.01 |
| Anxiety | 16577 (14.4) | 3519 (30.5) | <0.01 |
| Migraine | 4229 (3.67) | 868 (7.53) | <0.01 |
| Fibromyalgia | 27373 (23.8) | 5064 (43.9) | <0.01 |
| IBS | 6558 (5.69) | 1539 (13.4) | <0.01 |
| Asthma | 9008 (7.82) | 1446 (12.6) | <0.01 |
| Sleep apnea | 340 (0.29) | 99 (0.86) | <0.01 |
| Hyperthyroidism | 2774 (2.41) | 722 (6.26) | <0.01 |

Data expressed as case number (percentage)

pSS: Primary Sjögren syndrome; SD: standard deviation; CAD: coronary artery disease; DM: type 2 diabetes mellitus; CPP: chronic pelvic pain; CFS: chronic fatigue syndrome; IBS: irritable bowel syndrome

## Discussion

In this study, we showed that pSS patients exhibited significantly higher risks of bladder irritation symptoms compared with the control cohort using a large-scale, population-based database. Moreover, female gender, age younger than 65 years, and co-administration with treatment for sicca complex were associated with the development of OAB and BPS/IC. These novel findings suggest a possible pathogenic mechanism underlying the association of pSS and bladder irritation symptoms.

Previous study suggested that the rate of OAB was 56% in pSS patients and 22.7% in the comparison group.[3] We found that the risks of developing OAB in pSS patients was 1.68-fold compared with non-pSS patients. The possible mechanism of OAB development in

**Table 2. Risks of OAB and BPS/IC among pSS and comparison cohorts.**

|  | Comparison cohort | | | pSS cohort | | | Crude HR (95% CI) | Adjusted HR (95% CI) | p value |
|---|---|---|---|---|---|---|---|---|---|
|  | Event | PYs | Rate | Event | PYs | Rate |  |  |  |
| Total population | 1519 | 783936 | 19.4 | 352 | 76496 | 46.0 | 2.38(2.12–2.67) | 1.74(1.54–1.96) | <0.01 |
| OAB | 1443 | 784229 | 18.4 | 323 | 76645 | 42.1 | 2.29(2.03–2.59) | 1.68(1.48–1.91) | <0.01 |
| BPS/IC | 110 | 789256 | 1.39 | 39 | 77650 | 5.02 | 3.61(2.50–5.20) | 2.34(1.59–3.44) | <0.01 |

Adjusted for age, gender and comorbidities (hypertension, coronary artery disease, type 2 diabetes mellitus, chronic pelvic pain, chronic fatigue syndrome, depression, anxiety, migraine, fibromyalgia, urinary tract infection, irritable bowel syndrome, asthma, sleep apnea, hyperthyroidism)

PYs: person-years; Rate: per 10000 PYs; pSS: Primary Sjögren syndrome; OAB: overactive bladder disorder; BPS: bladder pain syndrome; IC: interstitial cystitis

CI: confidence interval; HR: hazard ratio by Cox proportional hazard model

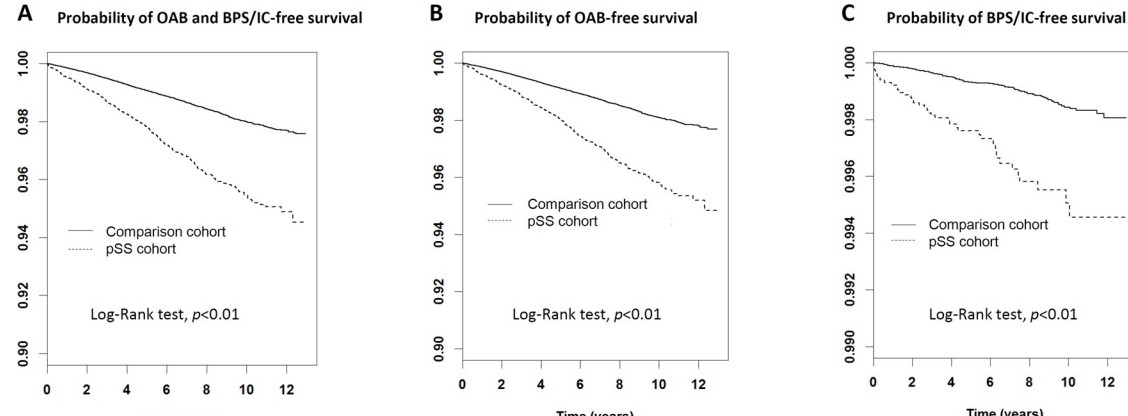

**Fig 2.** Comparisons of probabilities of (A) OAB and BPS/IC-free, (B) OAB-free, (C) BPS/IC-free survival between pSS and comparison cohorts. pSS: Primary Sjögren syndrome; OAB: overactive bladder; BPS: bladder pain syndrome; IC: interstitial cystitis.

pSS patients might be related to anti-M$_3$R IgG.[10] The M$_3$R is located not only in salivary glands and conjunctiva but also in the detrusor muscle layers of the urinary bladder. The cholinergic activity of M$_3$R has been reported to mediate contraction of smooth muscle in urinary bladder.[11] Wang et al. passively transferred inhibitory immunoglobulins with anti-M$_3$R activity from pSS patients into mice. An increased detrusor muscle contraction in response to cholinergic stimulation with increased expression of M$_3$R in the urinary bladder was observed. Therefore, cholinergic responses were enhanced during detrusor contraction and bladder

**Table 3. Risks of OAB and BPS/IC among SS and comparison cohorts by age and gender.**

| | | Comparison cohort | | pSS cohort | | Adjusted HR (95% CI) | *p* value |
|---|---|---|---|---|---|---|---|
| | | Event | Rate | Event | Rate | | |
| Age group* | | | | | | | |
| <65 | Total population | 727 | 11.8 | 201 | 33.4 | 1.82(1.55–2.15) | <0.01 |
| | OAB | 665 | 10.8 | 176 | 29.2 | 1.73(1.45–2.06) | <0.01 |
| | BPS/IC | 81 | 1.31 | 33 | 5.42 | 2.67(1.73–4.12) | <0.01 |
| ≥65 | Total population | 792 | 47.0 | 151 | 92.7 | 1.57(1.32–1.88) | <0.01 |
| | OAB | 778 | 46.1 | 147 | 90.2 | 1.56(1.31–1.88) | <0.01 |
| | BPS/IC | 29 | 1.69 | 6 | 3.57 | 1.39(0.56–3.43) | 0.48 |
| Gender ** | | | | | | | |
| Female | Total population | 1236 | 17.7 | 306 | 44.9 | 1.80(1.58–2.05) | <0.01 |
| | OAB | 1160 | 16.6 | 277 | 40.6 | 1.74(1.52–2.00) | <0.01 |
| | BPS/IC | 110 | 1.57 | 39 | 5.64 | 2.34(1.59–3.44) | <0.01 |
| Male | Total population | 283 | 33.1 | 46 | 55.0 | 1.33(0.96–1.83) | 0.09 |
| | OAB | 283 | 33.1 | 46 | 55.0 | 1.33(0.96–1.83) | 0.09 |
| | BPS/IC | 0 | 0 | 0 | 0 | - | - |

\* Adjusted for gender and comorbidities (hypertension, coronary artery disease, type 2 diabetes mellitus, chronic pelvic pain, chronic fatigue syndrome, depression, anxiety, migraine, fibromyalgia, urinary tract infection, irritable bowel syndrome, asthma, sleep apnea, hyperthyroidism)

\*\* Adjusted for age and comorbidities

Rate: per 10000 PYs; pSS: Primary Sjögren syndrome; OAB: overactive bladder disorder; BPS: bladder pain syndrome; IC: interstitial cystitis; CI: confidence interval; HR: hazard ratio by Cox proportional hazard model

**Table 4. Risks of OAB and BPS/IC by Sjogren syndrome-specific treatment.**

| Treatment | Total population | | OAB | | BPS/IC | |
|---|---|---|---|---|---|---|
| | Rate | aHR (95% CI) | Rate | aHR (95% CI) | Rate | aHR (95% CI) |
| Comparison cohort | 19.4 | reference | 18.4 | reference | 1.39 | reference |
| pSS cohort with | | | | | | |
| Dry mouth Tx* | 46.4 | 1.76(0.92–3.40) | 41.2 | 1.66(0.83–3.33) | 5.11 | 2.60(0.36–18.8) |
| Dry eyes Tx** | 37.4 | 1.43(1.20–1.71) | 34.9 | 1.40(1.16–1.68) | 4.32 | 2.13(1.24–3.64) |
| Both Tx | 59.6 | 2.16(1.84–2.53) | 53.4 | 2.06(1.74–2.43) | 6.91 | 2.93(1.81–4.74) |
| Others | 29.6 | 1.36(0.87–2.11) | 29.6 | 1.43(0.92–2.23) | 0 | - |

Adjusted for age, gender, and comorbidities (hypertension, coronary artery disease, type 2 diabetes mellitus, chronic pelvic pain, chronic fatigue syndrome, depression, anxiety, migraine, fibromyalgia, urinary tract infection, irritable bowel syndrome, asthma, sleep apnea, hyperthyroidism)

PYs: person-years; Rate: per 10000 PYs

Tx: treatment

*Dry mouth Tx: cevimeline, pilocarpine, and saliva substitute

**Dry eyes Tx: artificial tears

OAB: overactive bladder disorder; BPS: bladder pain syndrome; IC: interstitial cystitis; pSS: Primary Sjögren syndrome; CI: confidence interval; aHR: adjusted hazard ratio by Cox proportional hazard model

dilatation. Moreover, xerostomia, a cardinal symptom of pSS patients, may lead to excessive water intake, and consequently associated with OAB symptoms.[12]

In our study, we observed a 2.34-fold increased hazard ratio of BPS/IC in pSS patients. Van de Merve et al. also proposed that clinicians should be aware of an increased risk of BPS/IC in patients with pSS.[6] In addition, BPS/IC patients had relatively higher risks of developing pSS than the general population.[5] Anti-$M_3R$ IgG of pSS patients might also be crucial to the development of IC in both the early and late stages.[5] The detrusor muscle cells could produce interleukin(IL)-8, IL-6, and CCL5/RANTES in response to tumor necrosis factor(TNF)-α, and IL-1β.[13] In human airway smooth muscle cells, muscarinic receptor agonist may stimulate $M_3R$, leading to the secretion of IL-6 and CXCL8.[14] Similarly, an increased inflammatory cytokines secretion from detrusor muscle cells was observed after a compensatory increase in $M_3R$ expression caused by anti-$M_3R$ IgG. Moreover, In a murine model, the mast cells may induce IL-6, IL-8, and RANTES after the release of TNF-α and IL-1β.[15] Consequently, the increase of inflammatory cytokines induced by autoantibodies may drive mast cells to the bladder lamina propria of detrusor muscle, which is related to the pathogenesis of BPS/IC.[16]

It was reported that the mean age of BPS/IC among community-dwelling women who fulfilled the RAND BPS/IC epidemiology case definition was in their forties using various case definitions.[17] Moreover, pSS is prevalent in middle-aged females. Our study demonstrated that female pSS patients aged less than 65 years were at higher risks of OAB and BPS/IC compared with the geriatric group. In contrast, old age was associated with interstitial lung disease, another extra-glandular manifestation of pSS patients.[18] It is not clear whether anti-$M_3R$ IgG is differentially expressed in female SS patients of younger age. However, our findings suggest that bladder irritation symptoms were most severe in this group of pSS patients. Future study may be needed to elucidate the association of age, OAB, and BPS/IC in pSS patients.

Our results suggest that the risks of bladder irritation symptoms were increased in pSS patients with treatment for sicca complex. Although a previous study suggested that the extent of sicca symptoms in pSS patients was not related to serum IgG levels and the perceived ocular sicca symptoms were not related to tear production.[19] However, a recent study suggested that the presence of anti-Ro and La antibodies in tear fluid or serum was associated with the

severity of keratoconjunctivitis sicca.[20] Furthermore, anti-$M_3R$ expression was positively correlated with anti-Ro antibodies, but negatively correlated with saliva flow rate.[21] The association of pSS-specific treatment and bladder irritation in our study support the existence of a shared pathology between sicca symptoms and bladder disorders. Rheumatologists should question pSS patients with severe exocrine disorders to establish whether or not they have urinary symptoms.

This study had a number of strengths. First, we used an administrative database to conduct this nationwide, population-based cohort study. Thus, it may avoid selection bias in our investigation of the risks of bladder disorders in pSS patients with different ages, sexes, and specific treatments. In contrast, previous studies were largely relied on personal experiences or reviews of medical record in selected institutions with limited case numbers. The confounding effects of comorbidities cannot be avoided.[3,5,22] However, the outcome used in our study was defined by physicians' diagnosis. The more a person sees a physician, the more likely other diseases will be diagnosed. We cannot completely exclude substantial selection bias in this study. Second, complete information on outpatient, inpatient, and prescriptions were provided by the NHIRD, which meant that medical visits were not under-reported. Third, more than 98% of the population in Taiwan are of Chinese Han ethnicity; therefore it ensured that race was not a confounder. Fourth, the follow-up period was long enough for the manifestation of glandular dysfunction.[23,24] However, this study had several limitations. First, it was unable to assess treatment complications and the disease activities of pSS from the claims data. Second, the accuracy of diagnoses was a potential concern. Miscoding or misclassification of diseases may still have a chance to occur even though the BNHI regularly samples medical records to verify claims from hospitals. However, we believed that the accuracy of pSS diagnoses was still validated because the BNHI selects ≧ 2 well-trained and experienced rheumatologists to review patients' medical records and laboratory data for confirming pSS diagnoses before issuing a catastrophic illness certificate. Third, the prevalence of OAB and BPS/IC may have been underestimated. Berry SH et al. showed that the prevalence of BPS/IC in the community ranged from 2.7% to 6.5%.[17] However, only 0.46% of pSS patients and 0.19% of the comparison cohort in our study were classified as having OAB or BPS/IC. The discrepancies of prevalence rates might arise from different study designs and case definitions of questionnaires vs. claim database. We cannot exclude the possibility that the OAB or BPS/IC rates in healthy controls were also biased by under diagnosis. In view of the poor quality of life associated with OAB or BPS/IC [25] and available therapies, educational programs involving physicians and pSS patients would be helpful to minimize the knowledge gap. It is essential for rheumatologists to work closely with specialists in other medical fields to enhance case identification and treatment.

In conclusion, this nationwide, population-based study demonstrated an increased risk of bladder irritation disorders in pSS patients. Female gender, young age, and severe glandular dysfunction requiring treatment were potential risk factors. Greater awareness by clinicians and patients is imperative to avoid underdiagnosis of this treatable disease.

## Acknowledgments

We would like to thank the Biostatistics Task Force of Taichung Veterans General Hospital for their assistance in performing the statistical analysis.

## Author Contributions

**Conceptualization:** Chun-Kang Lee, Yi-Ming Chen.

**Data curation:** Ching-Pei Tsai, Ching-Heng Lin, Yi-Ming Chen.

**Formal analysis:** Ching-Heng Lin.

**Funding acquisition:** Yi-Ming Chen.

**Methodology:** Ching-Pei Tsai, Tsai-Ling Liao.

**Supervision:** Wen-Nan Huang, Yi-Hsing Chen.

**Writing – original draft:** Chun-Kang Lee.

**Writing – review & editing:** Ching-Pei Tsai, Tsai-Ling Liao, Wen-Nan Huang, Yi-Hsing Chen, Ching-Heng Lin, Yi-Ming Chen.

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
