## [Decision Letter · Decision Letter 0]

20 Sep 2019

PONE-D-19-18400

Overactive Bladder and Bladder Pain Syndrome/Interstitial Cystitis in Primary Sjögren’s Syndrome: A Nationwide Population-based Study

PLOS ONE

Dear Dr. Chen,

Thank you for submitting your manuscript to PLOS ONE. After careful consideration, we feel that it has merit but does not fully meet PLOS ONE’s publication criteria as it currently stands. Therefore, we invite you to submit a revised version of the manuscript that addresses the points raised during the review process.

Both of the reviewers think your experiment interesting, so please response the problems pointed out by them.

We would appreciate receiving your revised manuscript by Nov 04 2019 11:59PM. To enhance the reproducibility of your results, we recommend that if applicable you deposit your laboratory protocols in protocols.io, where a protocol can be assigned its own identifier (DOI) such that it can be cited independently in the future. For instructions see: http://journals.plos.org/plosone/s/submission-guidelines#loc-laboratory-protocols

We look forward to receiving your revised manuscript.

Kind regards,

Yoshiaki Taniyama, MD, PhD

Academic Editor

PLOS ONE

Journal Requirements:

Reviewers' comments:

Reviewer's Responses to Questions

**Comments to the Author**

1. Is the manuscript technically sound, and do the data support the conclusions?

Reviewer #1: Yes

Reviewer #2: Yes

2. Has the statistical analysis been performed appropriately and rigorously? 

Reviewer #1: I Don't Know

Reviewer #2: I Don't Know

3. Have the authors made all data underlying the findings in their manuscript fully available?

Reviewer #1: Yes

Reviewer #2: Yes

4. Is the manuscript presented in an intelligible fashion and written in standard English?

Reviewer #1: Yes

Reviewer #2: Yes

5. Review Comments to the Author

Reviewer #1: I commend the authors on a well written manuscript on an interesting finding. The high incidence of OAB and BPS/IC in women with Sjogren's is remarkable for sure.

I have several questions/comments for the authors:

1) Under 'Data Source' line 5 should read " Taiwan are included"

2) Can you further explain the decision to exclude subjects with diagnosis before 2001? Was the decision for newly diagnosed patients only to see progression more objectively?

3) In "Risks of OAB and BPS/IC" Why was the age of 65 years old chosen as a cut off?

4) In "Risks of OAB and BPS/IC" lines 12-14 does not make sense. I am not sure what you are trying to say here. Please rewrite.

5) In "Discussion" Why do you think rates of OAB BPS/IC were lower in your normal cohort population than described rates? Do you think it could be due to under diagnosis by physicians in the care of "normal" patients? Please explains.

All in all, a well written manuscript on an interesting subject.

Reviewer #2: Sjögren’s syndrome (SS) is an autoimmune disease affecting lacrimal and salivary glands. Primary SS has long been associated with interstitial cystitis/bladder pain syndrome (IC), but the extent of association remains unclear. Lee and colleagues used data from Taiwan’s National Health Insurance Research Database to examine the occurrence of IC and overactive bladder (OAB) among newly-diagnosed pSS patients over a 10-year period, relative to a non-pSS control group. Among 11,526 pSS patients, elevated hazard ratios were identified for both IC and OAB, and younger age, female gender, and sever glandular dysfunction were identified as risk factors.

The manuscript is generally well written, although tables should not be incorporated into the text, and page numbers would facilitate review.

Concerns:

1. On Page 15, minor editing should be applied to Lines 5-8. “…mast cells may induce…” and “drive mast cells to the bladder lamina propria…”

6. PLOS authors have the option to publish the peer review history of their article (what does this mean?). If published, this will include your full peer review and any attached files.

Reviewer #1: Yes: Sangeeta T Mahajan MD

Reviewer #2: Yes: Dr. David James Klumpp

---

## [Author Response · Author response to Decision Letter 0]

7 Oct 2019

Reply to the comments and suggestions from Reviewer 1

Thanks for your excellent review and useful comments

I commend the authors on a well written manuscript on an interesting finding. The high incidence of OAB and BPS/IC in women with Sjogren's is remarkable for sure. I have several questions/comments for the authors:

1) Under 'Data Source' line 5 should read " Taiwan are included"

We corrected the typo in the method section (page 5, line 5). 

2) Can you further explain the decision to exclude subjects with diagnosis before 2001? Was the decision for newly diagnosed patients only to see progression more objectively? 

In the Longitudinal Health Insurance Database (LHID) used in our study, missing data existed more frequently before 2000. Moreover, in order to exactly identify the index date to further analysis of hazard ratio for OAB and BPS/IC, we only enrolled newly diagnosed patients with primary Sjögren’s syndrome. We revised our method section (page 5, lines 19-20)

3) In "Risks of OAB and BPS/IC" Why was the age of 65 years old chosen as a cut off? 

The chronological age of 65 years is widely accepted as a definition of elderly in developed countries by many organizations including World Health Organization. Since Sjogren’s syndrome is an autoimmune disease involving female aged fifties to sixties and the mean age of pSS patients is our study was 53.8 ± 14.2 years, we decided to dichotomize age groups by 65 years in the analysis. 

4) In "Risks of OAB and BPS/IC" lines 12-14 does not make sense. I am not sure what you are trying to say here. Please rewrite. 

We revised our result section (page 9, lines 13-14)

5) In "Discussion" Why do you think rates of OAB BPS/IC were lower in your normal cohort population than described rates? Do you think it could be due to under diagnosis by physicians in the care of "normal" patients? Please explains. 

Previous studies of OAB and BPS/IC prevalence rates defined their cases mainly by questionnaires, while our study was based on claims database. We agreed with the reviewer’s opinion that there might be an underdiagnosis of OAB or BPS/IC by physicians in the non-pSS group. We revised our discussion section accordingly (page17, lines 1-4)

Thank you for your insightful suggestions.

Reply to the comments and suggestions from Reviewer 2

Thanks for your excellent review and useful comments

Sjögren’s syndrome (SS) is an autoimmune disease affecting lacrimal and salivary glands. Primary SS has long been associated with interstitial cystitis/bladder pain syndrome (IC), but the extent of association remains unclear. Lee and colleagues used data from Taiwan’s National Health Insurance Research Database to examine the occurrence of IC and overactive bladder (OAB) among newly-diagnosed pSS patients over a 10-year period, relative to a non-pSS control group. Among 11,526 pSS patients, elevated hazard ratios were identified for both IC and OAB, and younger age, female gender, and sever glandular dysfunction were identified as risk factors. The manuscript is generally well written, although tables should not be incorporated into the text, and page numbers would facilitate review. 

Concerns: 1. On Page 15, minor editing should be applied to Lines 5-8. “…mast cells may induce…” and “drive mast cells to the bladder lamina propria…”

Thanks for your valuable comments. We added page numbers to the manuscript and revised our discussion section. (page 15, lines 5 and 7)

---

## [Decision Letter · Decision Letter 1]

6 Nov 2019

Overactive Bladder and Bladder Pain Syndrome/Interstitial Cystitis in Primary Sjögren’s Syndrome: A Nationwide Population-based Study

PONE-D-19-18400R1

Dear Dr. Chen,

We are pleased to inform you that your manuscript has been judged scientifically suitable for publication and will be formally accepted for publication once it complies with all outstanding technical requirements.

With kind regards,

Yoshiaki Taniyama, MD, PhD

Academic Editor

PLOS ONE

Additional Editor Comments (optional):

Reviewers' comments:

Reviewer's Responses to Questions

**Comments to the Author**

1. If the authors have adequately addressed your comments raised in a previous round of review and you feel that this manuscript is now acceptable for publication, you may indicate that here to bypass the “Comments to the Author” section, enter your conflict of interest statement in the “Confidential to Editor” section, and submit your "Accept" recommendation.

Reviewer #1: All comments have been addressed

2. Is the manuscript technically sound, and do the data support the conclusions?

Reviewer #1: Yes

3. Has the statistical analysis been performed appropriately and rigorously? 

Reviewer #1: Yes

4. Have the authors made all data underlying the findings in their manuscript fully available?

Reviewer #1: Yes

5. Is the manuscript presented in an intelligible fashion and written in standard English?

Reviewer #1: Yes

6. Review Comments to the Author

Reviewer #1: Thank you for your comments and corrections. A very interesting manuscript and a good contribution to the medical literature.

7. PLOS authors have the option to publish the peer review history of their article (what does this mean?). If published, this will include your full peer review and any attached files.

Reviewer #1: Yes: Sangeeta T Mahajan MD

---

## [Editor Report · Acceptance letter]

12 Nov 2019

PONE-D-19-18400R1 

Overactive Bladder and Bladder Pain Syndrome/Interstitial Cystitis in Primary Sjögren’s Syndrome Patients: A Nationwide Population-based Study 

Dear Dr. Chen:

I am pleased to inform you that your manuscript has been deemed suitable for publication in PLOS ONE. Congratulations! Your manuscript is now with our production department. 

With kind regards,

on behalf of

Dr Yoshiaki Taniyama 

Academic Editor

PLOS ONE